# Nanoemulsions Containing *Mucuna pruriens* (L.) DC. Seed Extract for Cosmetic Applications

Suwaporn Chookiat [1], Tinnakorn Theansungnoen [2], Kanokwan Kiattisin [1,*] and Aekkhaluck Intharuksa [1,*]

1 Department of Pharmaceutical Sciences, Faculty of Pharmacy, Chiang Mai University, Chiang Mai 50200, Thailand; suwaporn_ch@cmu.ac.th
2 Green Cosmetic Technology Research Group, School of Cosmetic Science, Mae Fah Luang University, Chiang Rai 57100, Thailand; tinnakorn.the@mfu.ac.th
* Correspondence: kanokwan.k@cmu.ac.th (K.K.); aekkhaluck.int@cmu.ac.th (A.I.)

**Abstract:** *Mucuna pruriens* is a medicinal plant whose seeds have various types of pharmacological activities and are used in many traditional medicines. The aim of this study was to evaluate the phytochemicals as well as the anti-aging, antioxidant, and moisturizing properties of seed extracts of *M. pruriens* var. *pruriens* and *M. pruriens* var. *utilis*. In addition, the best extract was selected for the development of nanoemulsions. *M. pruriens* var. *utilis* had the highest total phenolic and total flavonoid contents. It had good antioxidant activity (the $IC_{50}$ of DPPH was 4.87 µg/mL, the FRAP value was 1.63 mg of $FeSO_4$/mg of extract, and the percentage of lipid peroxidation was 80.19%) and anti-aging activity (the percentages of inhibition of hyaluronidase, collagenase, and elastase were 26.41%, 51.16%, and 22.78%, respectively). The occlusive factor was 46.12 ± 1.72. Therefore, *M. pruriens* var. *utilis* seed extract was selected for the preparation of nanoemulsions. The results showed that the size, PDI, and zeta potential of nanoemulsions containing *M. pruriens* var. *utilis* seed extract at day 30 did not significantly differ from those at day 0. In addition, the %EE was 63.46%. A study of skin permeation showed that the retention in the membrane after six hours of skin permeation study was 44.19%. Therefore, nanoemulsions containing *M. pruriens* var. *utilis* seed extract have good potential for further use in cosmetic applications.

**Keywords:** *Mucuna pruriens*; antiaging; antioxidant; moisturizing property; nanoemulsion




## 1. Introduction

Skin aging is a process of skin deterioration caused by free radicals, environmental factors, and pollution [1]. Skin aging occurs due to two factors: extrinsic aging and intrinsic aging. Extrinsic aging is induced by external factors such as UV radiation, pollution, and smoking. Intrinsic aging, on the other hand, is a genetically determined and inevitable process that naturally causes physical changes in the skin. Intrinsic aging is determined by each person's genetics, and is affected by the degenerative effects of free radicals and the body's inability to repair their damage perfectly. It is a continuous process in which collagen production begins to slow, and elastin loses its spring. Dead skin cells are not shed as quickly, and the turnover of new skin cells decreases. Reactive oxygen species (ROSs) play an essential role in skin aging. Keratinocytes and fibroblasts are the leading producers of ROSs in the skin. When these ions are produced in large quantities, they are toxic and harmful to cells. The factors responsible for structural changes in the skin include decreased collagen and elastin, a slower turnover of dead skin cells, and high trans-epidermal water loss [2]. Ultraviolet (UV) radiation from sunlight increases the collagenase enzyme, which can degrade collagen in the skin [3]. Elastin can also be broken down by the elastase enzyme, which causes the loss of skin elasticity [4]. In addition, the loss of skin moisture causes the drying and aging of skin. Hyaluronic acid (HA) is a glycosaminoglycan that can hold water in the skin to keep it smooth and hydrated skin. The skin becomes dry and

wrinkles when HA is destroyed by hyaluronidase and free radicals [5]. Therefore, skin aging can be reduced by antioxidant and anti-aging agents [2].

In the quest for sources that possess antioxidant and anti-aging properties, natural extracts have gained considerable popularity as subjects for the investigation of these effects. *Mucuna pruriens* (L.) DC., commonly known as cowhage or the velvet bean plant, is a climbing shrub. The fruit resembles a long pea pod. The golden-brown hairs of this plant are toxic due to the presence of serotonin, causing itching upon contact. *M. pruriens* seeds contain various essential substances that are used as traditional medicines for several diseases [6]. In India and West Africa, this plant has been reported to have many medicinal properties, such as anti-Parkinson's activity [7], antitumor effects, anti-venom effects, antidiabetic effects, aphrodisiac effects [6], antioxidant activity [8], and antimicrobial activity [9]. As a result of these benefits, the Thai government has encouraged farmers to cultivate these plants for both domestic consumption and export. Thus, the cultivation of *M. pruriens* is extensive, and it encompasses both *M. pruriens* var. *pruriens* and *M. pruriens* var. *utilis* (Figure 1). Currently, there is a reduced demand for *M. pruriens* seeds, leading to an oversupply and consequent decline in their market price. Therefore, enhancing the value of *M. pruriens* seeds is a potential solution that would address this issue. The development of cosmetic products with extracts from *M. pruriens* seeds represents an additional means of enhancing the value of this plant. However, according to previous research, there are few studies on the cosmetic effects of *M. pruriens*. Therefore, research on the effects of cosmetics and the development of cosmetic products is necessary.

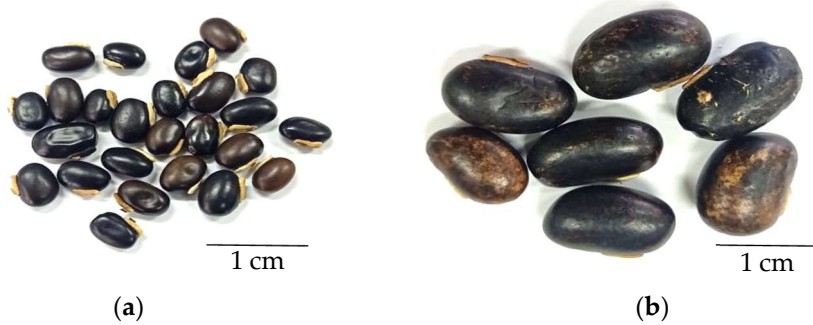

**Figure 1.** Seeds of *M. pruriens* var. *pruriens* (**a**) and *M. pruriens* var. *utilis* (**b**). The scale is 1 cm for each figure.

Lipid-based nanocarriers are a delivery system that is widely used in the cosmetic and cosmeceutical industries. They can protect an active ingredient from the environment [1]. Nanoemulsions are lipid-based nanocarriers consisting of oil, water, and a surfactant. The water and oil are mixed with an appropriate surfactant. The droplet size of nanoemulsions is 20–500 nm. As a result, they can penetrate through rough skin and are stable with respect to precipitation, inherent creaming, flocculation, coalescence, and sedimentation due to their small size. They can incorporate both hydrophilic and hydrophobic compounds. They are non-toxic, non-irritating, and small. They can be used to prepare a variety of formulas such as foams, creams, liquids, and sprays. These formulations are suitable for sensitive skin because they are non-toxic. Moreover, they can control and deliver an active compound to a target site on the skin. However, the stability of nanoemulsions depends on their pH, temperature, and Ostwald ripening [10]. The improved skin penetration of nanoemulsions depends on their nano-sized range and compositions [11].

This research focuses on promoting the utilization of *M. pruriens* seeds for cosmetic purposes. The aim of this study was to investigate the biological activities associated with the cosmetic properties of *M. pruriens* var. *pruriens*, in comparison with *M. pruriens* var. *utilis*. This comparison was carried out to identify the optimal extract for the development of a nano-cosmeceutical product. The chosen extract was integrated into nanoemulsions that were designed to deliver the extract to the deeper layers of the skin and enhance its stability.

## 2. Materials and Methods

### 2.1. Collection and Identification of Plant Materials

The seeds of two *Mucuna* species, namely, *M. pruriens* var. *utilis and M. pruriens var. pruriens*, were used in this study (Figure 1). Seeds of *M. pruriens* var. *utilis* were purchased from Chanthaburi province, whereas *M. pruriens* var. *pruriens* seeds were purchased from Phitsanulok province. The seeds of both species were sown and cultivated in the medicinal plant garden of the Faculty of Pharmacy, Chiang Mai University. The key morphological characteristics of the leaves, flowers, and pods were utilized for species identification by comparing them with specimens from the Herbarium at Queen Sirikit Botanical Garden, Chiang Mai, Thailand.

### 2.2. Chemicals

2,2-diphenyl-1-picrylhydrazyl (DPPH) was purchased from Fluka (Buchs, Switzerland). Folin–Ciocalteu reagent and 2,2′-azobis-(2-amidinopropane dihydrochloride) (AAPH) were purchased from Merck (Darmstadt, Germany). 6-hydroxy-2,5,7,8-tetramethyl chroman-2-carboxylic acid (Trolox), epigallocatechin-3-gallate (EGCG), gallic acid, linoleic acid, bovine serum albumin (BSA), L-3,4-dihydroxyphenylalanine (L-dopa), sodium chloride (NaCl), acetate buffer, ferrous chloride ($FeCl_2$), ferric chloride ($FeCl_3$), ferrous sulfate ($FeSO_4$), and sodium carbonate were purchased from Sigma-Aldrich (St. Louis, MO, USA). Ethanol, methanol, and dimethyl sulfoxide (DMSO) were purchased from Labscan Asia Co., Ltd. (Bangkok, Thailand). 2,4,6-Tris(2-pyridyl)-s-triazine (TPTZ) was purchased from Fluka Buchs (Buchs, Switzerland). Formic acid was purchased from Fisher Chemical (Loughborough, UK). Sorbitan oleate (Span® 80), polysorbate 80 (Tween® 80), and jojoba oil were purchased from Namsiang Co., Ltd. (Chiang Mai, Thailand). For enzyme and substrate substances, clostridium histolyticum collagenase (E.C.3.4.23.3), porcine pancreatic elastase (PE–E.C.3.4.21.36), bovine testis hyaluronidase (E.C.3.2.1.3.5), synthetic peptide of N-[3-(2-furyl) acryloyl]-Leu-Gly-Pro–Ala (FALGPA), N-succinyl-Ala–Ala–Ala–p-nitroanilide (AAAPVN), and hyaluronic acid were purchased from Sigma-Aldrich (St. Louis, MO, USA).

### 2.3. Preparation of M. pruriens Seed Extracts

The *M. pruriens* seeds of both varieties (var. *pruriens* and var. *utilis*) were washed with water before being dried. Next, the seeds were baked in a hot-air oven (Binder, Long Branch, New Jersey, USA) at approximately 50 °C for 6 h. Then, the dried seeds were mashed with a grinder until they became a powder. The preparation of *M. pruriens* seed extracts was adapted from the method of Maneepisamai et al. (2012) [12]. The powdered seeds were extracted with 95% *w/w* ethanol at a ratio of 10 g per 200 mL using a Soxhlet apparatus for 4 h. Then, the extract solution was concentrated using a rotary evaporator (EYELA, Tokyo, Japan). Subsequently, the weight of the crude extracts was calculated for the percentage yield (% yield).

$$\%yield = (weightofextract/dryweightofseeds) \times 100 \tag{1}$$

### 2.4. Determination of the Total Phenolic Content

The determination of the total phenolic content of each extract was conducted using a modified version of the method of Garzón et al. (2009) [13]. The *M. pruriens* seeds extracts were assessed for their total phenolic content using the Folin–Ciocalteu assay. Initially, the extract was dissolved in 50% *w/w* ethanol. Subsequently, the extract solution was mixed with 10% *v/v* Folin–Ciocalteu reagent in 1 mL of distilled water, followed by the addition of 4 mL of 7.5% *w/v* sodium carbonate. The resulting solution was incubated at room temperature (30 °C) in the dark for 30 min. The absorbance was measured at 765 nm using a UV–vis spectrophotometer (UV2600i, Shimadzu, Kyoto, Japan). The total phenolic content in the extract was calculated as gallic acid equivalents in mg of gallic acid/g of dry sample (GAE/g extract).

## 2.5. Determination of the Total Flavonoids Content

The determination of the total flavonoid content of each extract was conducted using a modified version of the method of Samatha et al. (2012) [14]. The *M. pruriens* seed extracts were mixed with 10 mL of distilled water and 0.3 mL of 5% *w/v* sodium nitrite, and then incubated at room temperature (30 °C) for 5 min. Then, the extract solution was combined with 0.15 mL of 10% *w/v* aluminum chloride and left at room temperature (30 °C) for 15 min. The absorbance was measured at 765 nm using a UV–vis spectrophotometer (UV2600i, Shimadzu, Kyoto, Japan). The total flavonoid content in the extract was calculated as quercetin equivalent in mg of quercetin/g of dry sample (QE/g extract).

## 2.6. Chemical Marker Analysis of M. pruriens Seed Extracts with High-Performance Liquid Chromatography (HPLC)

The L-dopa in the *M. pruriens* seeds extracts was analyzed using HPLC (Shimadzu, Kyoto, Japan). The conditions under which HPLC was carried out were set according to Theansungnoen et al. (2022) and Dhanani et al. (2015) [15,16]. The concentrations of L-dopa were 170, 125, 62.5, 31.25, and 15.625 ppm for the standard curve. The crude extracts (250 ppm) were dissolved in absolute methanol. A C18 column (30 m × 0.25 mm × 0.25 μm) (HP-5MS column, Agilent Technologies, Santa Clara, California, USA) was used as the stationary phase. The injection volume for all samples was 10 μL. The mobile phase for the gradient consisted of 0.1% *w/w* formic acid in deionized water (pH 3.75) and absolute methanol. The ratios of the gradient are shown in Table 1. The extracts and L-dopa were eluted with a flow rate of 1.0 mL/min. The absorbance was detected at 285 nm. The concentration of L-dopa in the extracts was calculated using a standard curve for L-dopa.

**Table 1.** The ratios of the HPLC gradient.

| Time (s) | Value (%) | |
| --- | --- | --- |
| | Formic Acid | Absolute Methanol |
| 0.01 | 50 | 50 |
| 1.45 | 2 | 98 |
| 1.75 | 30 | 70 |

## 2.7. Fatty Acids Analysis with Gas Chromatography–Mass Spectrometry (GC–MS)

For the analysis of volatile phytochemicals in the extracts, we followed a method of separation that was previously described in Theansungnoen's study [15]. This method involved using gas chromatography (Agilent 19091S-433, Agilent Technologies, Santa Clara, CA, USA) with a GC column (30 m × 0.25 mm × 0.25 μm) (HP-5MS column, Agilent Technologies, Santa Clara, CA, USA). The column oven temperature was initially set to 60 °C, and then ramped up to 290 °C over a 10-minute period. Subsequently, the extracts were diluted with absolute methanol and introduced into the system using the split sampling method at a 1:10 ratio, with helium gas serving as the sample carrier at a flow rate of 1 mL/min at 290 °C. The mass spectrum was recorded for each sample after a solvent-cutting time of 3 min, and the analysis lasted for 33 min. The GC–MS software (MassHunter GC/MS Acquisition 10.0.0368) was used to compute the retention time (RT), and to correct the peak areas in each spectrum. Phytochemicals were identified by matching the retention time of eluted peaks in the GC column with the mass spectra (MS) through comparison with the NIST and WILEY library databases. This analysis facilitated the identification of the primary compounds present in the extracts.

## 2.8. Determination of Antioxidant Activity

### 2.8.1. 1,1-Diphenyl-2-Picrylhydrazyl (DPPH) Radical Scavenging Assay

The DPPH radical scavenging assay was adapted from Wan et al. (2015) [17]. First, 0.1 mg/mL of extract solution was diluted with 50% *w/w* ethanol. Then, 20 μL of the extract solution was mixed with 180 μL of 2,2-diphenyl-1-picrylhydrazyl (DPPH) solution

in a 96-well plate, and incubated in the dark at room temperature (30 °C) for 30 min. The absorbance was measured at 517 nm using a microplate reader (BMG Labtech, Aylesbury, UK). The results were compared with those of Trolox and gallic acid. The percentage of inhibition was calculated with the following equation:

$$\%\text{inhibition} = [(\text{Control} - \text{Blank of control}) - (\text{Sample} - \text{Blank of sample})/\text{Control} - \text{Blank of control}] \times 100 \quad (2)$$

where "Control" represents the absorbance of the mixture in the absence of the extract solution, "Blank of control" pertains to the absorbance of the solvent, "Sample" refers to the absorbance of the extract solution, and "Blank of the sample" corresponds to the absorbance of the extract solution without 2,2-diphenyl-1-picrylhydrazyl (DPPH).

### 2.8.2. Ferric-Reducing Antioxidant Power (FRAP) Assay

The FRAP reagent was formulated through the combination of a 300 mM acetate buffer at a pH of 3.6, 5 mL of 10 mM 2,4,6-Tris(2-pyridyl)-s-triazine (TPTZ) dissolved in 40 mM of 37% $v/v$ HCl, and 5 mL of 20 mM ferric chloride solution. Then, 1 mg/mL of extract solution was dissolved in 50% $w/w$ ethanol and thoroughly mixed with 180 μL of FRAP reagent. Subsequently, the absorbance was determined at 593 nm using a microplate reader (BMG Labtech, Aylesbury, UK). The results were quantified as FRAP values and compared with those of Trolox and gallic acid [18].

### 2.8.3. Lipid Peroxidation Inhibition According to the Ferric Thiocyanate Assay

The ferric thiocyanate assay for lipid peroxidation inhibition was adapted from Olszewska (2011) [19]. The extract solution was dissolved in 50% $v/v$ dimethyl sulfoxide (DMSO). Subsequently, 150 μL of the extract solution was mixed with 350 μL of PBS (pH 7.0), 100 μL of DI water, and 350 μL of 2.4% $w/w$ linoleic acid in a test tube. Then, 50 μL of 2,2-azobis(2-amidinopropane) dihydrochloride (AAPH) was added, and the mixture was incubated in the dark at 50 °C. Additionally, 5 μL of the sample was combined with 5 μL of 20 mM ferrous chloride in 3.5% $w/v$ hydrochloric acid, 5 μL of 10% $w/w$ ammonium thiocyanate, and 185 μL of 75% $v/v$ methanol, and this mixture was incubated at room temperature (30 °C) for 3 min. The absorbance was recorded at 500 nm using a microplate reader (BMG Labtech, Aylesbury, UK). The results were compared with those of Trolox and gallic acid. The percentage of inhibition was calculated using the following equation:

$$\%\text{inhibition} = [(\text{Control} - \text{Blank of control}) - (\text{Sample} - \text{Blank of sample})/(\text{Control} - \text{Blank of control})] \times 100 \quad (3)$$

In this context, "Control" signifies the absorbance of the mixture lacking the extract solution, "Blank of control" refers to the absorbance of the solvent, "Sample" denotes the absorbance of the extract solution, and "Blank of sample" represents the absorbance of the extract solution in the absence of ammonium thiocyanate and ferrous chloride.

### 2.9. Determination of Anti-Aging Activity

#### 2.9.1. Anti-Hyaluronidase Activity According to a Turbidimetric Assay

The turbidimetric assay for anti-hyaluronidase activity was adapted from Be Tu and Tawata, 2015 [20]. First, 5 mg/mL of extract solution was dissolved in 20% $w/w$ dimethyl sulfoxide (DMSO). Subsequently, 50 μL of extract solution was mixed with 100 μL of hyaluronidase enzyme and incubated at 37.5 °C for 10 min in a water bath. Following this, 100 μL of 0.03% $w/v$ hyaluronic acid in PBS (pH 5.35) was added to the extract solution, and incubated at 37.5 °C for 45 min in a water bath. Then, 1 mL of bovine serum albumin acid solution was added and incubated at room temperature (30 °C) for 10 min. The absorbance was measured at 450 nm using a microplate reader (BMG Labtech, Aylesbury, UK). The results were compared with those of gallic acid and tannic acid. The percentage of inhibition was calculated with the following equation:

$$\%\text{inhibition} = [(A - B)/A] \times 100 \quad (4)$$

where "A" represents the absorbance of the sample, "B" pertains to the absorbance of the control.

2.9.2. Anti-Collagenase Activity According to a Spectrophotometric Assay

The spectrophotometric assay for anti-collagenase activity was modified from Thring et al., 2017 [21]. First, 5 mg/mL of extract solution was dissolved in 20% *w/w* dimethyl sulfoxide (DMSO). Next, 10 μL of the extract solution was mixed with 40 μL of collagenase enzyme in tricine buffer (pH 7.5). Then, 50 μL of 2 mM N-[3-(2-furyl) acryloyl]-Leu-Gly-Pro–Ala (FALGPA) was added to the extract solution. The absorbance was measured at 335 nm for 20 min in kinetic mode using a microplate reader (BMG Labtech, Aylesbury, UK). The results were compared with those of gallic acid and epigallocatechin gallate (EGCG). The percentage of inhibition was calculated using the following equation:

$$\%Collagenase\,inhibition = ((A - B)/A) \times 100 \tag{5}$$

where "A" represents the reaction rate of collagenase and the substrate, "B" represents the reaction rate of the extract solution, collagenase, and the substrate.

2.9.3. Anti-Elastase Activity According to Spectrophotometric Assay

The spectrophotometric assay for anti-elastase activity was adapted from Kim et al., 2004 [22]. First, 5 mg/mL of extract solution was dissolved in 20% *w/w* dimethyl sulfoxide (DMSO). Subsequently, 50 μL of the extract solution was mixed with 25 μL of 4.4 mM N-succinyl-Ala–Ala–Ala–p-nitroanilide (AAAVPN) in Tris-HCL buffer (pH 8.0). Following this, 25 μL of elastase in Tris-HCL buffer was added to the extract solution. The absorbance was measured at 450 nm in the kinetic mode using a microplate reader (BMG Labtech, Aylesbury, UK). The results were compared with those of gallic acid and epigallocatechin gallate (EGCG). The percentage of inhibition was calculated with the following equation:

$$\%Elastase\,inhibition = [(A - B)/A] \times 100 \tag{6}$$

where "A" represents the reaction rate of elastase, Tris-HCl buffer, and the substrate, "B" represents the reaction rate of the extract solution, elastase, Tris-HCl buffer, and the substrate.

*2.10. Determination of the Moisturizing Properties with an Occlusion Assay*

The determination of the moisturizing properties of each extract was carried out using an occlusion assay, which was adapted from Da Silva et al. (2020) [23]. Initially, 30 mL of DI water was prepared in a beaker and covered with cellulose filter paper (Whatman, Maidstone, Kent, UK). Subsequently, 1 mL of the sample of 1 mg/mL in 50% *w/w* ethanol was dispensed onto the cellulose filter paper. The sample was placed in a hot-air oven (Binder, Long Branch, NJ, USA) at 40 °C for 6, 24, and 48 h. After this incubation period, the weight of the DI water in the beaker was compared at different time intervals. Naked cellulose filter paper was used as a blank. The occlusion factor (F) was calculated using the following equation:

$$F = [(A - B)/A] \times 100 \tag{7}$$

where "A" represents the value of water loss from the beaker without a sample, "B" represents the value of water loss from the beaker with the sample.

*2.11. Development of Unloaded Nanoemulsions*

2.11.1. Preparation of Unloaded Nanoemulsions

The preparation of unloaded nanoemulsions in this study was adapted from the method of Salvia-Trujillo et al. (2013) [24]. Jojoba oil was chosen for use in the oil phase (fix at 5% *w/w*). Polysorbate 80 (Tween® 80) and sorbitan oleate (Span® 80) were used as emulsifiers at a ratio of 1:1, and their concentrations were varied as follows: 5, 10, and 15 %

*w/w* (Table 2). The oil and water phases were prepared separately at room temperature (30 °C). Then, both phases were combined using a high-shear homogenizer (T 25 digital ULTRA-TURRAX®, IKA, Königswinter, Germany) operating at 6000 rpm for a duration of 10 min to generate the coarse emulsions. Following this stage, the coarse emulsions were further processed to create nanoemulsions through ultrasonication (Vernon Hills, IL, USA) by applying 60% amplitudes for 10 min.

**Table 2.** Compositions of unloaded nanoemulsions.

| Formulation | Jojoba Oil (% *w/w*) | Tween® 80 to Span® in Rato of 1:1 (% *w/w*) | Deionized Water (% *w/w*) |
|---|---|---|---|
| 1 | 5 | 5 (2.5:2.5) | 90 |
| 2 | 5 | 10 (5:5) | 85 |
| 3 | 5 | 15 (7.5:7.5) | 80 |

### 2.11.2. Characterization and Stability Testing of Unloaded Nanoemulsions

The physical characteristics of the nanoemulsions, including their particle size, zeta potential, and polydispersity index (PDI), were analyzed using the technique of dynamic light scattering (DLS) (Zetasizer, Malvern ZS, Malvern Panalytical Ltd., Malvern, UK) [20]. The sample was diluted with deionized water at a ratio of 10:90. In addition, the pH of the formulations was analyzed using a pH meter (Starter 2100, OHAUS, Parsippany, NJ, USA). Furthermore, the stability of these nanoemulsions was assessed at room temperature (30 °C) for 30 days [25].

### 2.12. Development of Nanoemulsions Containing M. pruriens var. utilis Seeds Extract

#### 2.12.1. Preparation of Nanoemulsions Containing *M. pruriens* var. *utilis* Seeds Extract

A 0.05% *w/w* concentration of the *M. pruriens* var. *utilis* seed extract was chosen for incorporation into the nanoemulsions due to its potent biological activity. The extract was dissolved in the oil phase (jojoba oil). Then, the formulation process followed the nanoemulsions preparation procedure described in Section 2.11.1.

#### 2.12.2. Characterization and Stability Study of Nanoemulsions Containing *M. pruriens* var. *utilis* Seed Extract

The nanoemulsions containing *M. pruriens* var. *utilis* seed extract were kept at 30 °C for 30 days and underwent 6 heating–cooling cycles (the formulation was kept at 45 °C for 48 h, and then moved to 4 °C for 48 h as one cycle) to study their stability [26]. After the stability study, their physical properties—particle size, zeta potential, and polydispersity index (PDI)—were analyzed using the technique of dynamic light scattering (DLS) (Zetasizer, Malvern ZS, Malvern Panalytical Ltd, Malvern, UK), as mentioned above in Section 2.11.2.

#### 2.12.3. Determination of Entrapment Efficiency

The entrapment efficiency of the nanoemulsions containing 0.05% *w/w* of *M. pruriens* var. *utilis* seeds extract was determined using an indirect method. The amount of L-dopa with entrapment is referred to as $W_{total}$, and amount of L-dopa without entrapment was analyzed as $W_{free}$. $W_{total}$ was analyzed using 1 mL in a tube containing 0.5 mL of nanoemulsion and 0.5 mL of 95% ethanol. $W_{free}$ was analyzed using 1 mL in a filter tube containing 0.5 mL of nanoemulsion. This involved centrifuging the nanoemulsions at 10,000 rpm at room temperature (30 °C) for 30 min. Subsequently, the quantity of the active compound present in the supernatant was analyzed using HPLC (Shimadzu, Kyoto, Japan) [25]. The percentage of entrapment efficiency (%EE) was calculated using the following formula:

$$\text{Entrapment efficiency (\%EE)} = [(W_{total} - W_{free})/W_{total}] \times 100 \tag{8}$$

where $W_{total}$ represents the amount of all of the extract in the nanoemulsions, and $W_{free}$ represents the amount of free extract that was not entrapped in the nanoemulsions.

2.12.4. Skin Retention Study of Nanoemulsions Containing *M. pruriens* var. *utilis* Seed Extract

The skin retention study was conducted with modifications of the method described by Junyaprasert et al., 2009 [27]. Franz diffusion cells were used in this study. A Strat® M membrane (Merck KgaA, Darmstadt, Germany) was placed in the donor chamber. Phosphate buffer at a pH of 5.5 was used as receiver medium, and was maintained at $32 \pm 2$ °C with continuous stirring facilitated by a magnetic bar. Then, 2 mL of the sample was added to the donor chamber. After 6 h, the Strat® M membrane (Merck KgaA, Darmstadt, Germany) was rinsed with phosphate buffer to eliminate any excess formulation. Subsequently, the membrane was cut into small pieces, submerged in absolute methanol, and sonicated for 15 min. The amount of extract that permeated into the skin was analyzed using HPLC (Shimadzu, Kyoto, Japan).

*2.13. Statistical Analysis*

All of the results are expressed as the mean and standard deviation (SD). Statistical analyses were performed using a paired *t*-test and one-way ANOVA with multiple comparisons while employing Tukey's test. Significance was considered to be at the level of $p < 0.05$.

## 3. Results

*3.1. M. pruriens Seed Extracts*

The physical characteristics of *M. pruriens* var. *utilis* seeds extract indicated that it was a red-brown semisolid extract with a pungent odor. On the other hand, the *M. pruriens* var. *pruriens* seeds extract was a green-brown semisolid extract with a characteristic odor. The percentage yields of *M. pruriens* var. *utilis* and *M. pruriens* var. *pruriens* were 8.30% and 9.14%, respectively (Table 3).

**Table 3.** Percentage yield of seeds extracts of *M. pruriens* var. *utilis* and *M. pruriens* var. *pruriens*.

| Type | %yield |
|---|---|
| *M. pruriens* var. *utilis* | 8.30[a] |
| *M. pruriens* var. *pruriens* | 9.14[a] |

Different letters in the column indicate significant difference at $p < 0.05$.

*3.2. Total Phenolic and Total Flavonoid Content of the M. pruriens Seed Extracts*

The total phenolic contents of the *M. pruriens* var. *utilis* and *M. pruriens* var. *pruriens* seed extracts were $252.96 \pm 3.27$ gallic acid/g extract and $223.54 \pm 9.33$ mg gallic acid/g extract, respectively, as shown in Figure 2. The total flavonoid contents of the seed extracts from *M. pruriens* var. *utilis* and *M. pruriens* var. *pruriens* were $734.82 \pm 28.48$ mg quercetin/g extract and $696.74 \pm 37.30$ mg quercetin/g extract, as shown in Figure 2.

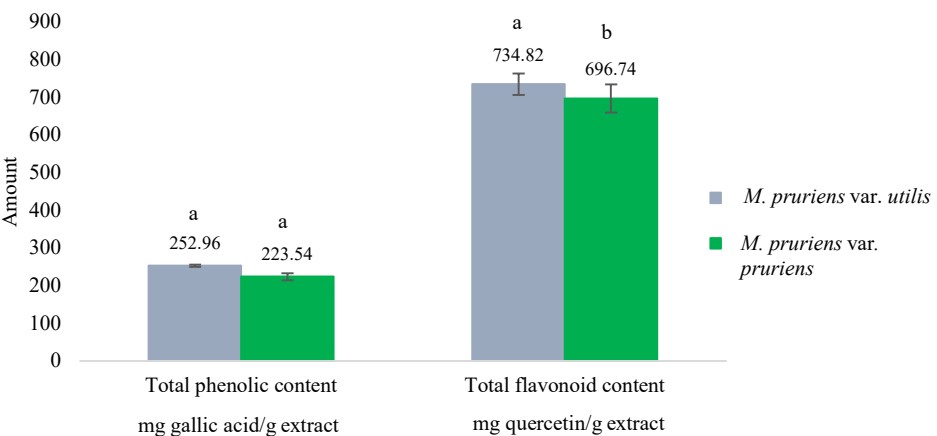

**Figure 2.** Total phenolic and total flavonoid contents of seed extracts from *M. pruriens* var. *utilis* and *M. pruriens* var. *pruriens* (Different letters are significantly different at $p < 0.05$ when analyzed via one-way ANOVA).

*3.3. Chemical Marker Analysis of the M. pruriens Seed Extracts with High-Performance Liquid Chromatography (HPLC)*

High-performance liquid chromatography (HPLC) was used to analyze the amount of L-dopa, a major phytochemical substance in *Mucuna* seeds. The calibration curve was very straight, with a correlation of 0.9993, as shown in Figure 3. The tallest peak in the fingerprint of the HPLC chromatogram is depicted in Figure 4. The amounts of L-dopa in the *M. pruriens* var. *utilis* and *M. pruriens* var. *pruriens* extracts were 0.36 mg/ mg extract and 0.17 mg/ mg extract, respectively.

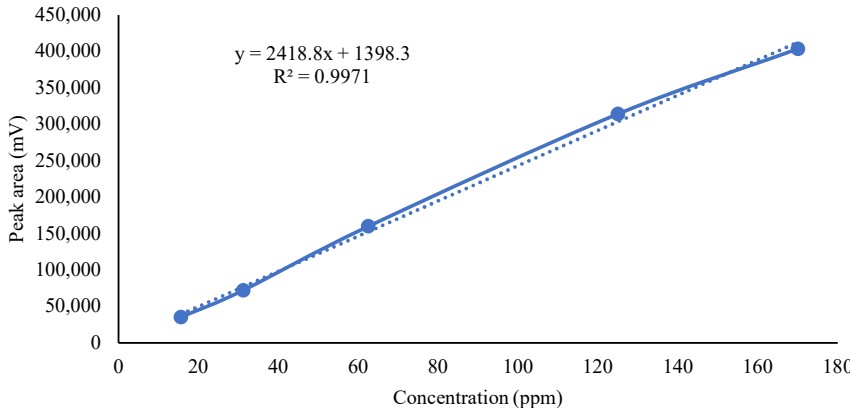

**Figure 3.** Standard curve of L-dopa from five concentrations detected by HPLC.

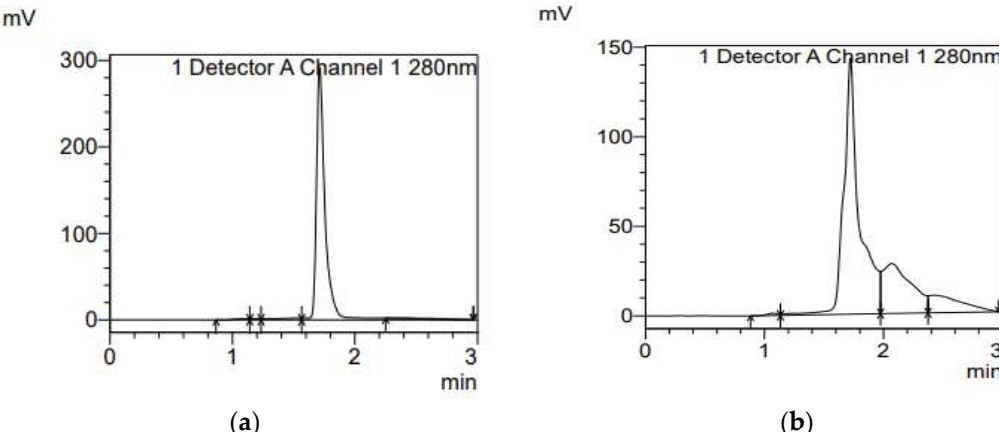

**Figure 4.** *Cont.*

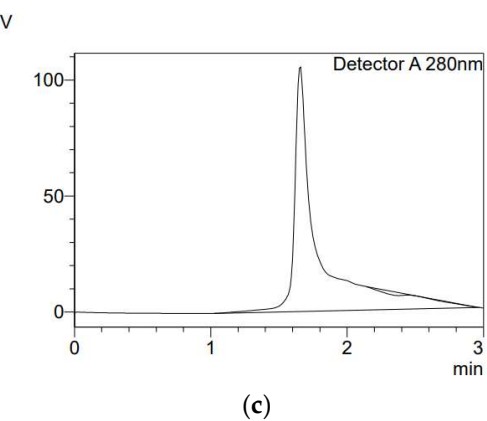

(**c**)

**Figure 4.** HPLC chromatograms of L-dopa from standard L-dopa (**a**), *M. pruriens* var. *utilis* seeds extract (**b**), and *M. pruriens* var. *pruriens* seeds extract (**c**).

*3.4. Fatty Acids Analysis with Gas Chromatography–Mass Spectrometry (GC–MS)*

The fatty acids in the ethanolic extracts of both *M. pruriens* samples (*M. pruriens* var. *utilis* and *M. pruriens* var. *pruriens*) were examined using the GC–MS technique. The GC chromatograms of the *M. pruriens* var. *utilis* and *M. pruriens* var. *pruriens* extract samples are presented in Figure 5a,b, respectively. The mass spectra in each chromatogram were identified by comparing them with the NIST and WILEY library databases, and the phytochemical constituents in each *M. pruriens* seed extract were correlated and are displayed in the figures. In general, both samples exhibited similar results in terms of phytochemical substances, with almost identical retention times (RTs) for compounds such as 2,4-Dit-butylphenyl 5-hydroxypentanoate (RT = 12.39 min), isopropyl myristate (RT = 15.86 min), 9(Z)-octadecenamide (RT = 20.91 min), octadecanamide (RT = 21.10 min), and oxirane, tetradecyl (RT = 22.57 min). However, one distinct chemical compound was found to differ between the two samples (RT = 19.32 min), and their mass spectra are shown in Figure 6. The *M. pruriens* var. *pruriens* contained nonanamide (Figure 6a), while the *M. pruriens* var. *utilis* contained hexadecanamide (Figure 6b).

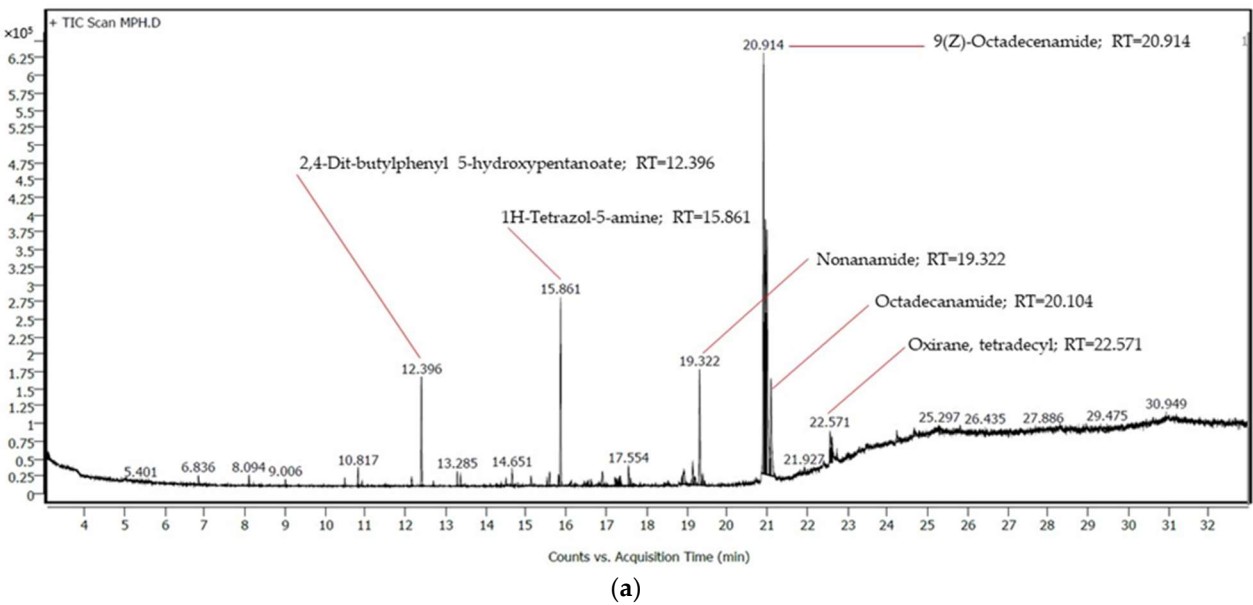

(**a**)

**Figure 5.** *Cont.*

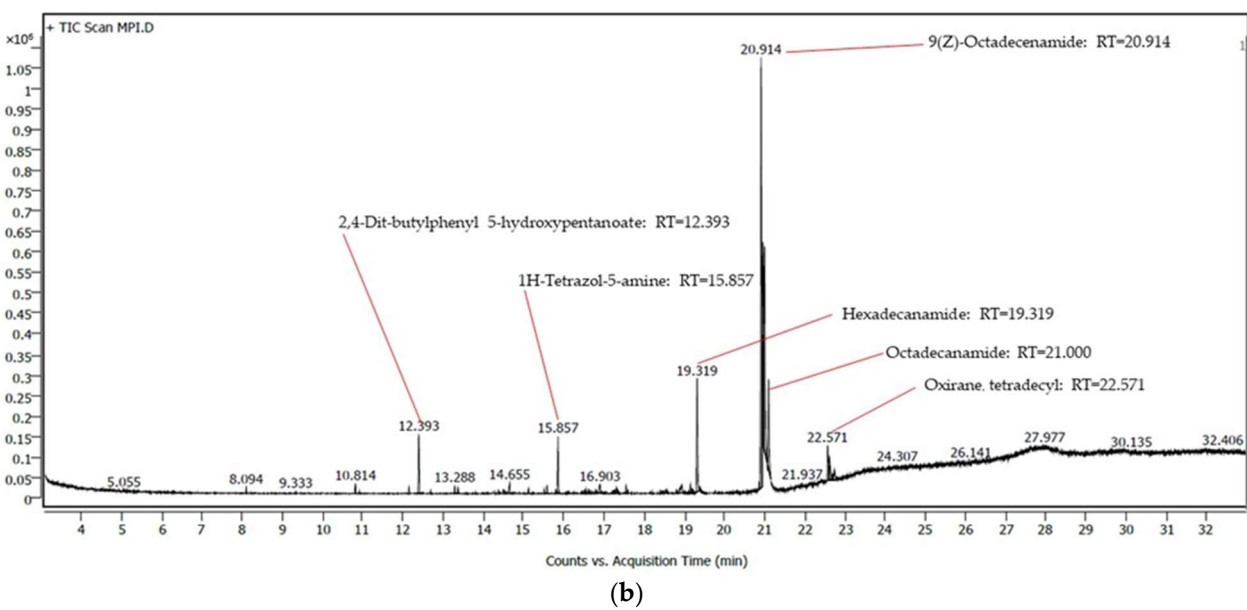

**Figure 5.** GC–MS chromatograms of *Mucuna* seeds extracts *M. pruriens* var. *utilis* (**a**) and *M. pruriens* var. *pruriens* (**b**).

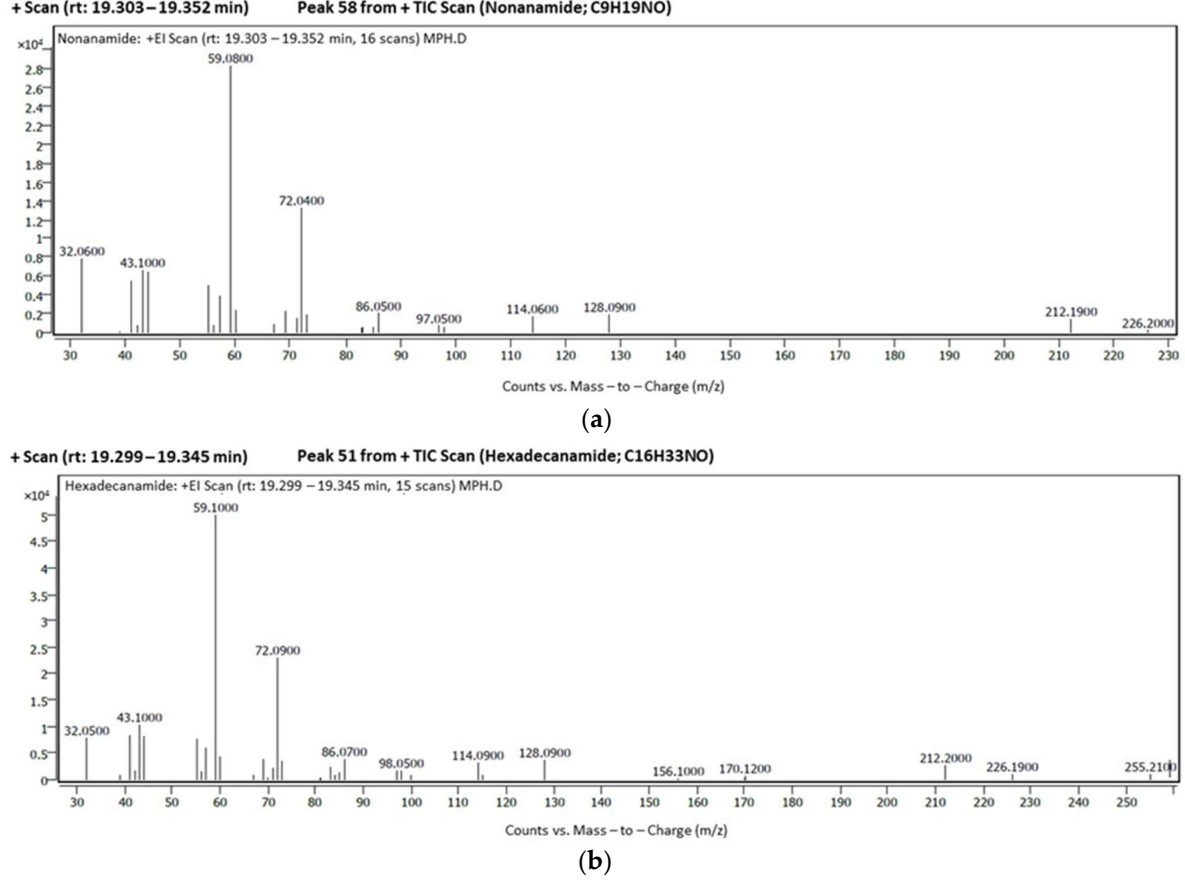

**Figure 6.** The mass spectra in each chromatogram of *Mucuna* seeds extracts *M. pruriens* var. *utilis* (**a**) and *M. pruriens* var. *pruriens* (**b**).

*3.5. Antioxidant Activities of the Mucuna Seeds Extracts*

3.5.1. 2,2-diphenyl-1-picrylhydrazyl (DPPH) Radical Scavenging Assay

The anti-radical activities of the extracts from *M. pruriens* var. *utilis* and *M. pruriens* var. *pruriens* were evaluated using the 2,2-diphenyl-1-picrylhydrazyl (DPPH) method. The results show in Figure 7. The $IC_{50}$ values of the *M. pruriens* var. *utilis* and *M. pruriens* var. *pruriens* extracts were 4.87 ± 0.01 and 47.67 ± 0.03, respectively. The $IC_{50}$ values of gallic acid and Trolox were 1.72 ± 0.02 and 1.17 ± 0.01, respectively.

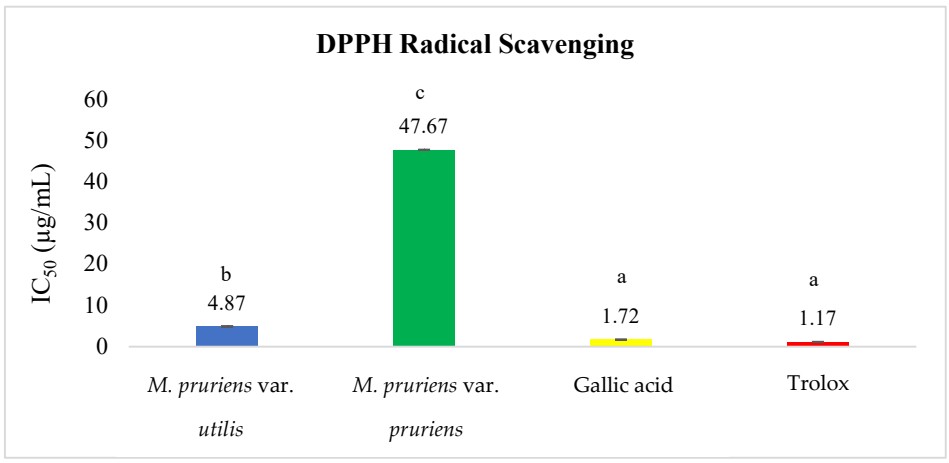

**Figure 7.** $IC_{50}$ values of the extracts from *M. pruriens* seeds (*M. pruriens* var. *utilis* and *M. pruriens* var. *pruriens*) and the standard compounds, gallic acid and Trolox, when tested by the DPPH method (a, b, and c are significantly different at $p < 0.05$ when analyzed by one-way ANOVA).

3.5.2. Ferric-Reducing Antioxidant Power (FRAP) Assay

The abilities of the *M. pruriens* seed extracts from the two *Mucuna* species (*M. pruriens* var. *utilis* and *M. pruriens* var. *pruriens*) to reduce ferric ions using the FRAP method as shown in Figure 8. The FRAP values of *M. pruriens* var. *utilis* and *M. pruriens* var. *pruriens* were 0.66 ± 0.01 and 0.14 ± 0.01 mg $FeSO_4$/mg extract, respectively. The FRAP values of gallic acid and Trolox were 0.36 ± 0.05 and 0.55 ± 0.05 mg $FeSO_4$/mg standard, respectively.

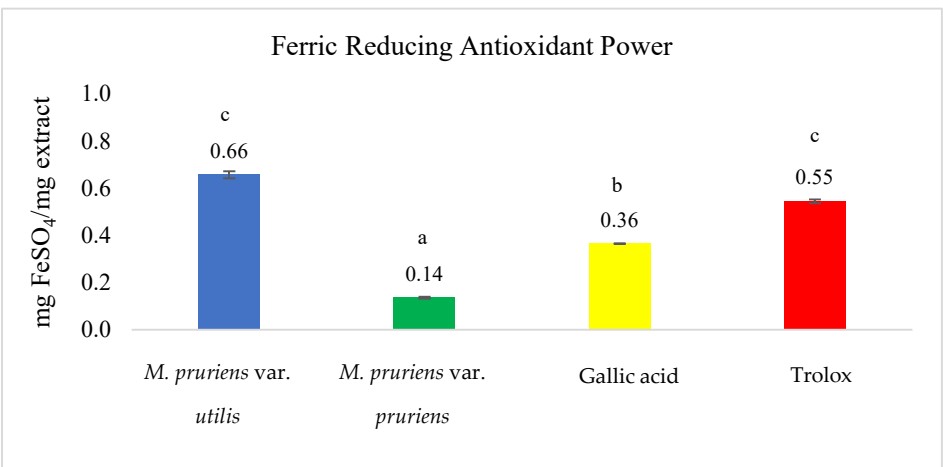

**Figure 8.** The FRAP values of the extracts from *M. pruriens* seeds (*M. pruriens* var. *utilis* and *M. pruriens* var. *pruriens*) and the standard compounds, gallic acid and Trolox, when tested by the FRAP method (a, b, and c are significantly different at $p < 0.05$ when analyzed by one-way ANOVA).

### 3.5.3. Lipid Peroxidation Inhibition According to the Ferric Thiocyanate Assay

The inhibitory effect of the *M. pruriens* seed extracts on lipid peroxidation was evaluated using the ferric thiocyanate assay as shown in Figure 9. The extracts from the two *Mucuna* species (*M. pruriens* var. *utilis* and *M. pruriens* var. *pruriens*) at a concentration of 1 mg/mL had percentage inhibition values of 80.19 ± 0.01 and 75.82 ± 0.02%, respectively. The values of gallic acid and Trolox at 1 mg/mL were 60.61 ± 0.01 and 75.60 ± 0.92%, respectively.

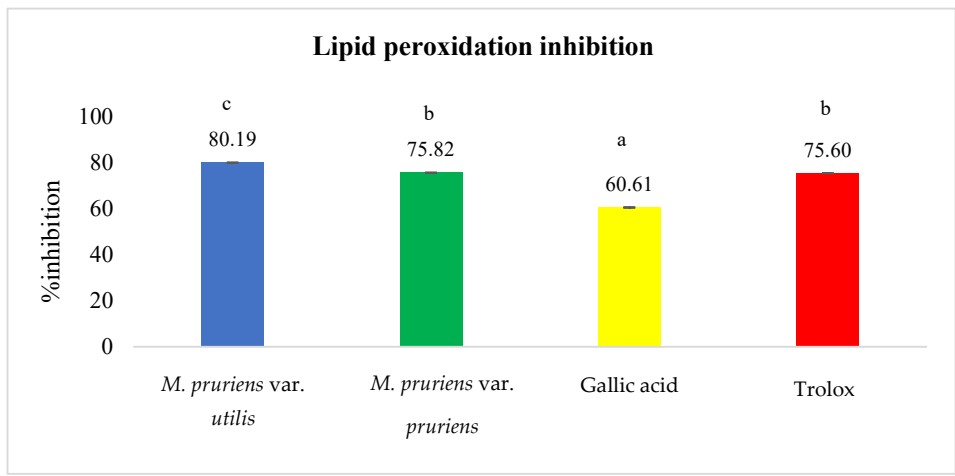

**Figure 9.** The percentage inhibition of the extracts from *M. pruriens* seeds (*M. pruriens* var. *utilis* and *M. pruriens* var. *pruriens*) and the standard compounds, gallic acid and Trolox, when tested by lipid peroxidation inhibition assay (a, b, and c are significantly different at $p < 0.05$ when analyzed by one-way ANOVA).

### 3.6. Anti-Aging Activities of Mucuna Seeds Extracts

### 3.6.1. Anti-Hyaluronidase Activity According to the Turbidimetric Assay

The anti-hyaluronidase activity of *M. pruriens* seed extracts from the two species (*M. pruriens* var. *utilis* and *M. pruriens* var. *pruriens*) was evaluated using a turbidimetric assay as shown in Figure 10. The concentration of the extracts was 0.2 mg/mL. The percentage inhibition values of *M. pruriens* var. *utilis* and *M. pruriens* var. *pruriens* were 26.41 ± 4.96% and 12.49 ± 1.52%, respectively. The percentage inhibition values for the standard compounds, gallic acid and tannic acid, were 1.20 ± 0.38% and 80.62 ± 0.25%.

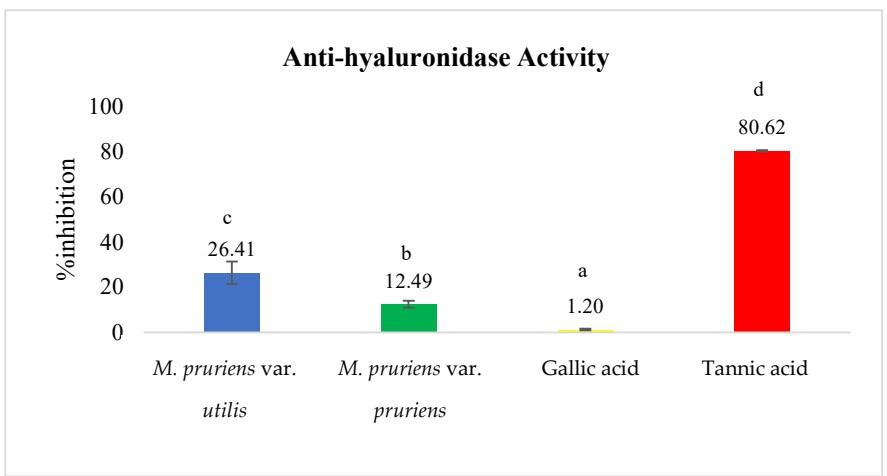

**Figure 10.** The percentage inhibition values of the extracts from *M. pruriens* seeds (*M. pruriens* var. *utilis* and *M. pruriens* var. *pruriens*) and the standard compounds (gallic acid and tannic). a, b, c, and d are significantly different at $p < 0.05$ when analyzed by one-way ANOVA.

### 3.6.2. Anti-Collagenase Activity According to the Spectrophotometric Assay

The anti-collagenase activity of *M. pruriens* seed extracts from both *Mucuna* species (*M. pruriens* var. *utilis* and *M. pruriens* var. *pruriens*) was evaluated using a spectrophotometric assay. The results are shown in Figure 11. The concentration of the extracts was 0.45 mg/mL. The percentage inhibition values were 51.16 ± 4.96 and 40.70 ± 5.81% for *M. pruriens* var. *utilis* and *M. pruriens* var. *pruriens*, respectively. The percentage inhibition value for the standard compound, epigallocatechin gallate (EGCG), was 84.30 ± 0.58%.

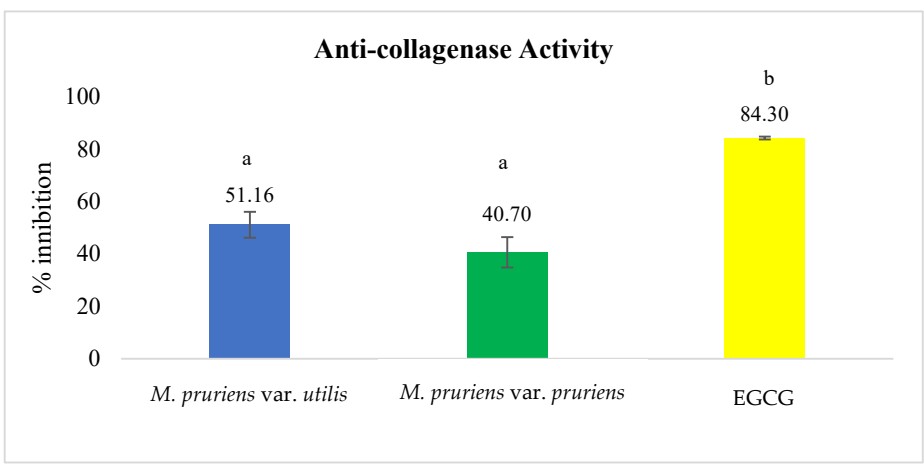

**Figure 11.** The percentage inhibition values of the extracts from *M. pruriens* seeds (*M. pruriens* var. *utilis* and *M. pruriens* var. *pruriens*) and epigallocatechin gallate (EGCG). a and b are significantly different at $p < 0.05$ when analyzed by one-way ANOVA.

### 3.6.3. Anti-Elastase Activity by Spectrophotometric Assay

The anti-elastase activity of *M. pruriens* seed extracts from the two species (*M. pruriens* var. *utilis* and *M. pruriens* var. *pruriens*) was evaluated using a spectrophotometric assay. The concentration of the extracts was 1.25 mg/mL. The results are shown in Figure 12. The percentage inhibition values were 22.78 ± 3.85 and 6.20 ± 3.96% for *M. pruriens* var. *utilis* and *M. pruriens* var. *pruriens*, respectively. The percentage inhibition value for the standard compound, epigallocatechin gallate (EGCG), was 88.98 ± 3.53%.

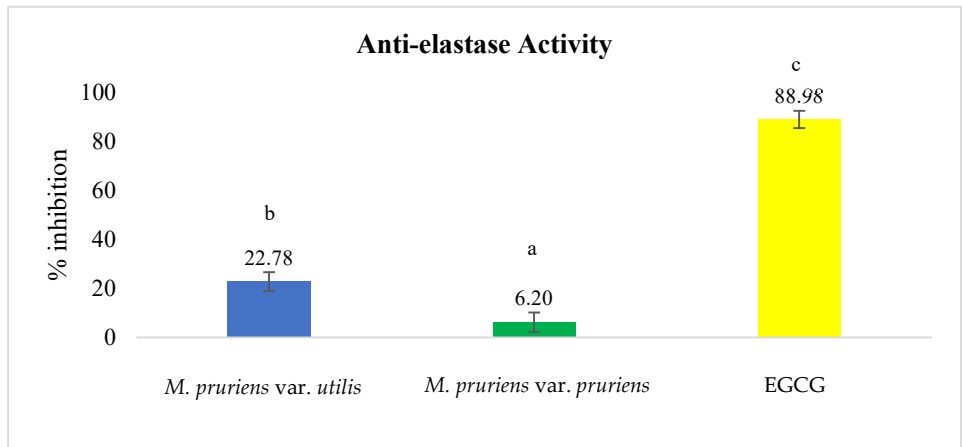

**Figure 12.** The percentage inhibition values of the extracts from *M. pruriens* seeds (*M. pruriens* var. *utilis* and *M. pruriens* var. *pruriens*) and epigallocatechin gallate (EGCG). a, b, and c are significantly different at $p < 0.05$ when analyzed by one-way ANOVA.

### 3.7. Moisturizing Properties of Mucuna Seeds Extracts According to the Occlusion Assay

The moisturizing properties of *M. pruriens* seeds at 1 mg/mL (*M. pruriens* var. *utilis* and *M. pruriens* var. *pruriens*) were analyzed using an occlusion assay. The measurements were performed at three time points (6, 24, and 48 h). The results are shown in Figure 13. At 6 h, the occlusive factors (F) were 5.99 ± 0.47 and 6.09 ± 0.77, respectively. At 24 h, the occlusive factors (F) were 20.74 ± 0.89 and 20.89 ± 1.40, respectively. At 48 h, the occlusive factors (F) were 46.12 ± 1.72 and 46.53 ± 1.63, respectively.

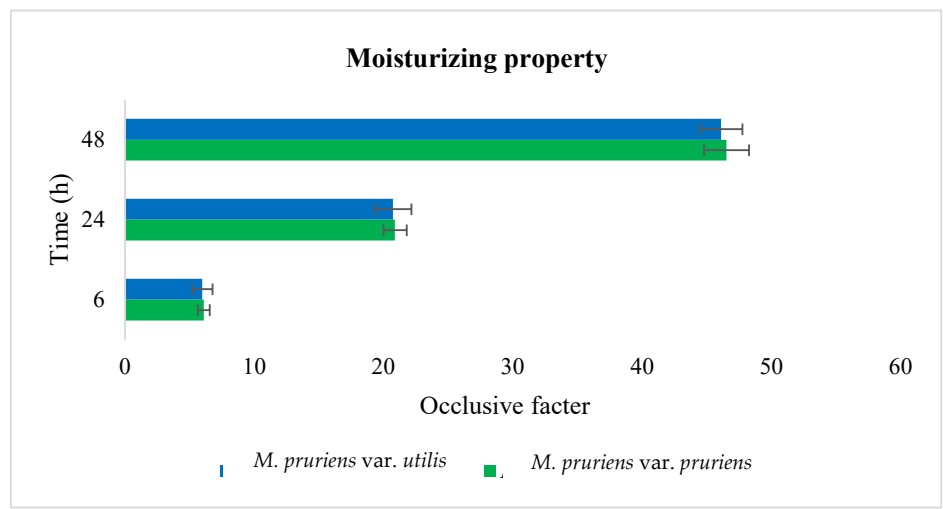

**Figure 13.** Occlusive factors of *M. pruriens* seeds extracts from the 2 varieties (*M. pruriens* var. *utilis* and *M. pruriens* var. *pruriens*) at a concentration of 1 mg/mL.

### 3.8. Development of Unloaded Nanoemulsions

3.8.1. Characterization and Stability of Unloaded Nanoemulsions

Nanoemulsions were prepared using a high-shear homogenizer and ultrasonication. Jojoba oil was fixed at 5% *w/w*. Polysorbate 80 (Tween® 80) and sorbitan oleate (Span® 80) were used as emulsifiers at a ratio of 1:1, and their concentrations were verified: 5, 10, and 15 % *w/w* (Table 3). The particle size, PDI, and zeta potential of all of the formulations are shown in Figure 14. All formulations presented small particle sizes in the range of 100–120 nm, with a narrow range for the PDI and a high zeta potential. The pH values of all formulations were in the range of 5.5–6. Formulation 2 was selected for the stability analysis because it had the smallest particle size. Formulation 2 was kept at room temperature (30 °C) for 30 days. The results are presented in Figure 15. The results showed that the particle size, polydispersity index (PDI), zeta potential, and pH value of the selected unloaded nanoemulsions were not significantly different when compared with those at day 0.

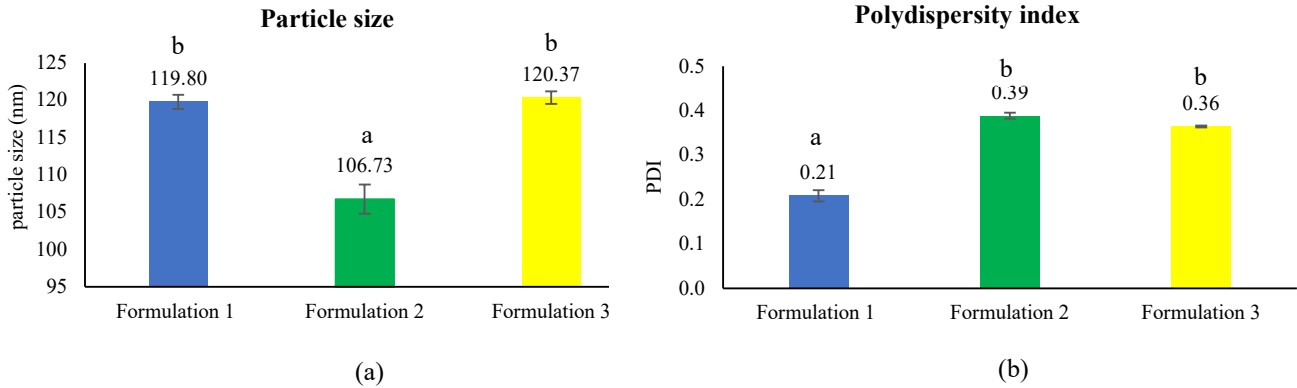

(a)                                                                 (b)

**Figure 14.** *Cont.*

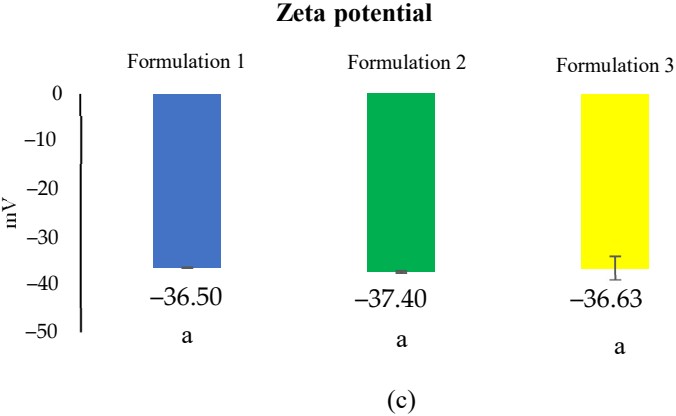

**Figure 14.** Particle size (**a**), polydispersity index (PDI) (**b**), and zeta potential (**c**) of unloaded nanoemulsions (a and b are significantly different at $p < 0.05$ when analyzed by one-way ANOVA).

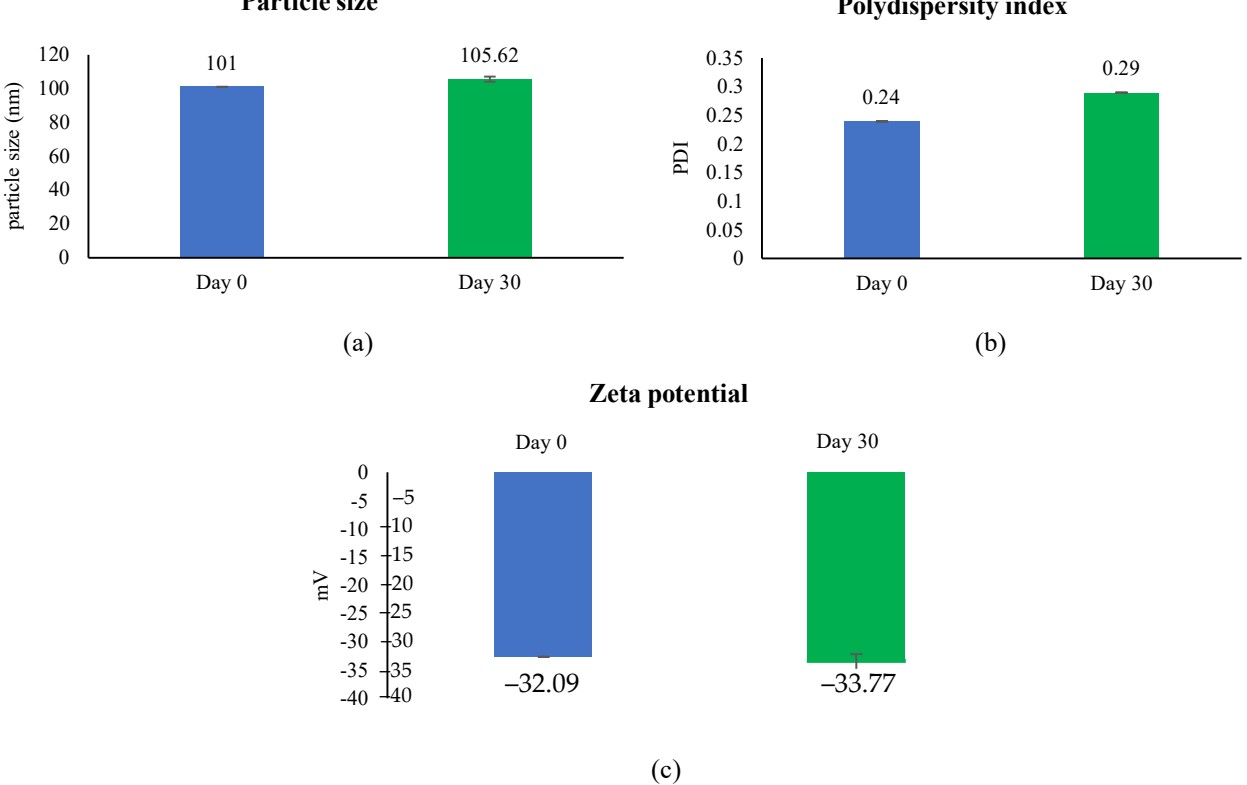

**Figure 15.** Particle size (**a**), polydispersity index (PDI) (**b**), and zeta potential (**c**) of selected unloaded nanoemulsions before and after stability test at 30 days.

3.8.2. Characterization of Nanoemulsions Containing the *M. pruriens* var. *utilis* seed extract

The results concerning the antioxidant, anti-aging, and moisturizing properties showed that M. pruriens var. utilis had better biological activity than that of M. pruriens var. pruriens. Therefore, M. pruriens var. utilis was selected for loading in the nanoemulsion formulation. The nanoemulsion containing the *M. pruriens* var. *utilis* seed extract was composed of three main components: 5% *w/w* jojoba oil, 5% *w/w* polysorbate 80 (Tween® 80), 5% *w/w* sorbitan oleate (Span® 80), and an aqueous phase. The best concentration of the M. pruriens var. utilis seed extract for the nanoemulsion was 0.05 mg/mL, which was the optimal concentration for making efficient nanoemulsions. The results of particle size, PDI, and zeta potential of the formulation after preparation are shown in Figure 16. The results showed that the particle size, PDI, and zeta potential of nanoemulsions containing

the *M. pruriens* var. *utilis* seeds extract were 149.9 nm, 0.26, and −32.69 mV, respectively. In addition, the nanoemulsions showed good stability after storage at room temperature (30 °C) for 1 month and six heating–cooling cycles. The droplet size, PDI, and zeta potential did not change after the stability study as shown in Figure 16. The pH value of the nanoemulsion containing the *M. pruriens* var. *utilis* seeds extract was 5.23, and did not change after storage for 1 month and after six heating–cooling cycles. The physical appearance of unloaded nanoemulsion and nanoemulsion containing *M. pruriens* var. *utilis* seeds extract is shown in Figure 17.

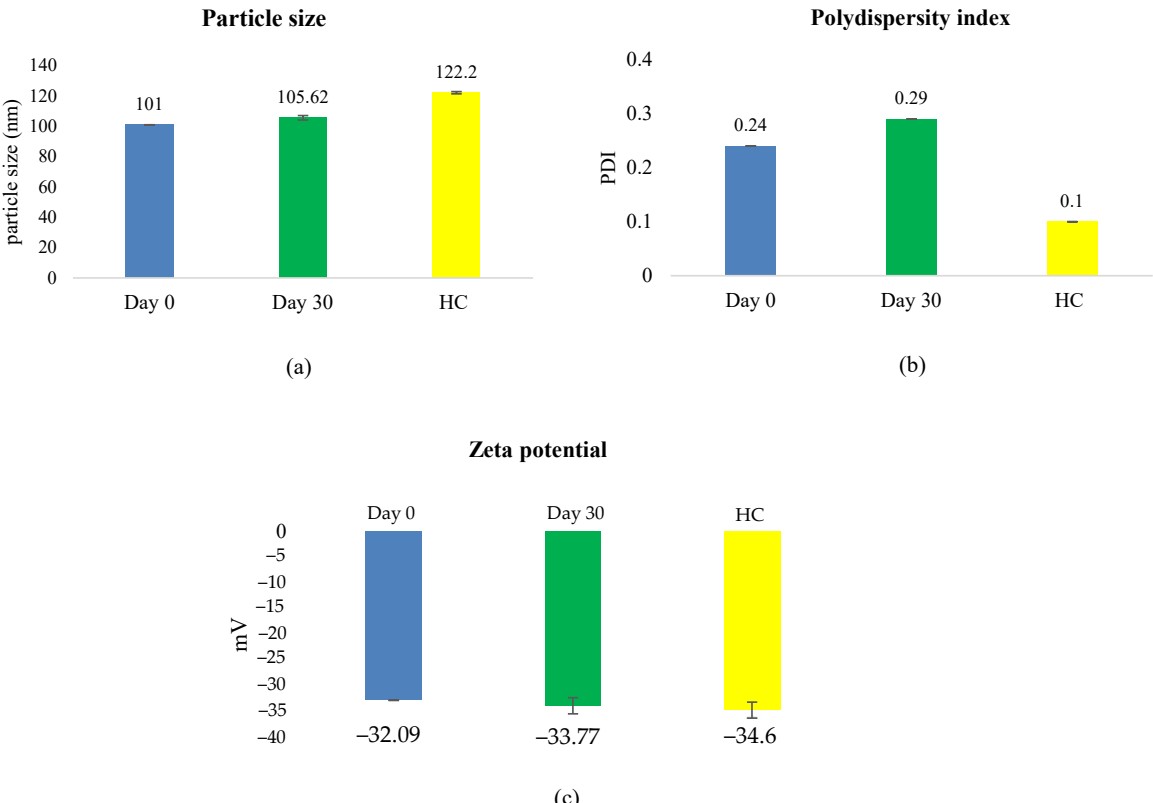

**Figure 16.** Particle size (**a**), polydispersity index (PDI) (**b**), and zeta potential (**c**) of nanoemulsions containing *M. pruriens* var. *utilis* seeds extract before and after stability study for 30 days and heating–cooling for 6 cycles (* is significantly different at $p < 0.05$ when analyzed by paired sample *t*-test).

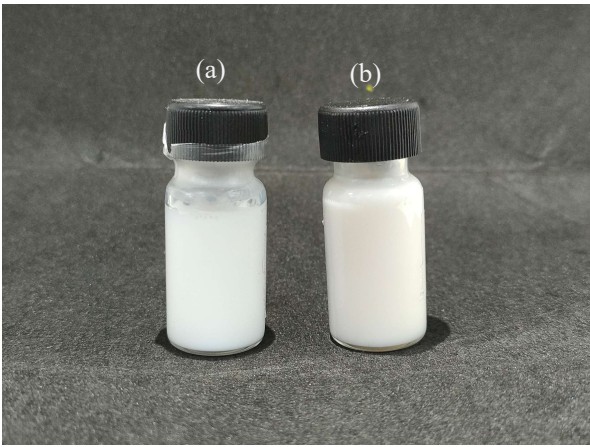

**Figure 17.** Characteristics of unloaded nanoemulsion (**a**) and nanoemulsion containing *M. pruriens* var. *utilis* seeds extract (**b**).

### 3.8.3. Entrapment Efficiency of Nanoemulsions Containing *M. pruriens* var. *utilis* Seeds Extract

The entrapment efficiency can be used to evaluate the efficiency of substance encapsulation in a nanoemulsion delivery system. It was found that the entrapment efficiency of the nanoemulsions containing the *M. pruriens* var. *utilis* seeds extract was 63.46%.

### 3.8.4. Skin Retention Study of Nanoemulsions Containing *M. pruriens* Seeds Extract

The percentage of L-dopa remaining in the Strat® M membrane of the nanoemulsions containing the *M. pruriens* var. *utilis* seeds extract after six hours was 44.19 ± 1.69%.

## 4. Discussion

The physical characteristics of the two varieties of *M. pruriens* are different; the seeds of *M. pruriens* var. *utilis* measure approximately 1 cm, and they are round and black. The seed of *M. pruriens* var. *pruriens* are similar to those of *M. pruriens* var. *utilis*, but they are more petite, approximately 0.5 cm long. Soxhlet extraction is a suitable method for extracting *M. pruriens* seeds because the active ingredients in them (gallic acid, caffeic acid, p-coumaric acid, luteolin, epicatechin, and kaempferol) are heat resistant [28]. The advantages of this method are that it saves time and solvent. In terms of solvent selection, the active ingredients in *M. pruriens* seeds are polar compounds, and extraction requires the maximum amount of plant-derived active ingredients. Therefore, 95% ethanol was used as the extraction solvent, as it was effective in extracting the active ingredients of the plant. The same extraction solvent that was chosen in this study was chosen in the research of Aware et al. (2019). In their study, it was found that using 95% *v/v* ethanol for extraction from *M. pruriens* seeds increased the antioxidant capacity, total phenolic content, and total flavonoid content [29]. The *M. pruriens* seeds extracts consisted of phenolic and flavonoid compounds, which was consistent with the research of Divya et al. (2017) [6] and Njemuva et al. (2019) [30], who found phenolic compounds in Mucuna. Examples of phenolic compounds found in *M. pruriens* include gallic acid, caffeic acid, and p-coumaric acid [30]. In this case, *M. pruriens* var. *utilis* seeds had a greater total phenolic content than that of *M. pruriens* var. *pruriens* seeds. This may have been due to the plant cultivation source, storage, and other control variables [31]. The flavonoids found in *M. pruriens* include luteolin, epicatechin, and kaempferol [32].

An antioxidant agent is a substance that can inhibit or delay an oxidation reaction; which occurs due to free radicals. It is a stable substance that can prevent or slow down oxidation caused by free radicals. It can inhibit oxidation in foods, quench free radicals, and stop oxidation chains in vivo [33]. Such molecules can donate an electron to a free radical, reducing its damage capacity (Lobo et al., 2010) [34]. Plant-derived antioxidants include carotenoids, vitamins, phenols, flavonoids, dietary glutathione, and endogenous metabolites [33]. Generally, there are many methods of determining the antioxidant activities of plant extracts. In this research, there was a focus on 2,2-diphenyl-1-picrylhydrazyl (DPPH), ferric-reducing antioxidant power (FRAP), and lipid peroxidation inhibition assays, which explained the different antioxidant mechanisms. The DPPH radical scavenging assay was performed to evaluate the anti-radical properties of the extracts. In this case, a DPPH radical is a synthetic substance in the form of a stable purple radical. The FRAP assay works on the principle that antioxidants can donate electrons to a complex $\left[ \text{Fe (III)(TPTZ)}^2 \right]^{3+}$, causing it to change into $\left[ \text{Fe(III)(TPTZ)}^2 \right]^{2+}$. The amount of $\left[ \text{Fe(III)(TPTZ)}^2 \right]^{+}$ that is formed can be used to estimate the reducing ability in terms of FRAP value units. Lipid peroxidation is caused by a chain reaction of oxidation of unsaturated fatty acids, which produces many lipid peroxides. This leads to changes in the cell membrane, affecting various enzymes that function within it. A reaction is initiated by the formation of free radicals, which then react to form lipid radicals [35]. The anti-radical activity of seed extracts from *M. pruriens* var. *utilis* was higher than that of seed extracts from *M. pruriens*

var. *pruriens* when evaluated with the DPPH method. Interestingly, *M. pruriens* var. *utilis* had better anti-radical activity against DPPH radicals than that of gallic acid and Trolox. The results of this study were consistent with the chemical compositions of the extracts, which showed high levels of phenolic and flavonoid compounds. These two compounds have strong antioxidant properties. The results were also compatible with those of the study of Divya et al. (2017) [6], who found that *Mucuna* seed extracts presented antioxidant properties by removing superoxide and hydroxyl radicals. In the FRAP assay, the FRAP value can be used to estimate the ability of an extract to reduce metal, which is a key for stimulating an oxidation reaction. The extracts with higher FRAP values showed better reducing properties. When comparing the extracts from the two *Mucuna* species, *M. pruriens* var. *utilis* was shown to have a higher FRAP value than that of *M. pruriens* var. *pruriens*. In addition, the FRAP value of the *M. pruriens* var. *utilis* extract was not significantly different from the standard. This result indicated that the *M. pruriens* var. *utilis* extract could effectively donate an electron to the complex. This was consistent with the findings of Jimoh et al. (2020) [31] and Rajeshwar et al. (2005) [36], who reported that *M. pruriens* seed extracts had antioxidant activity in the FRAP assay. The lipid peroxidation inhibition assay is related to a chain reaction of unsaturated fatty acids, which produce many lipid peroxides. These lipid peroxides can cause changes in the cell membrane and affect the enzymes that function within it. The initial reaction is the formation of free radicals, which then react with lipids to form lipid radicals. The extracts from the two *Mucuna* varieties had lipid peroxidation inhibition activity exceeding 50%. Compared with the standard compounds, the percentage inhibition values were significantly different, indicating that the two varieties had a high ability to stop chain reactions of lipid radicals. In their research, Theansungnoen et al. (2022) [15] and Divya et al. (2017) [6] also found that extracts from *M. pruriens* seeds had the ability to inhibit lipid peroxidation. In addition, the antioxidant activities increased with increasing extract concentration.

Hyaluronidase is an enzyme that degrades hyaluronic acid, which has the property of retaining skin moisture. Extracts with inhibitory activity react with hyaluronidase in a competitive inhibition mode. This reduces the ability of hyaluronidase to bind to hyaluronic acid, resulting in fewer or no reactions. This allows hyaluronic acid to remain in the system and react with acetic albumin to form a white residue. In this study, the *M. pruriens* var. *utilis* seed extract showed the highest inhibitory activity. Collagenase is an enzyme that can degrade collagen, a protein that helps bind cells in the skin, tendons, joints, ligaments, blood vessels, and organs. It also helps support the structure of the skin. In addition, collagen plays a crucial role in skin repair after injury. Extracts with anti-collagenase activity can inhibit the breakdown of collagen. The kinetic mode was used to test the anti-collagenase activity of the extracts. This method involved measuring the rate of enzyme-catalyzed reactions. As time passed, the extract competed with the substrate to bind with the enzyme. When no more substrate was available, the reaction rate reached a constant value, and then decreased over time in an inverse proportional manner. As time increased, the measured absorbance decreased. The results showed that the extracts could only partially bind to the enzyme, allowing some enzymes to still react with the substrate. In comparison, the positive control, epigallocatechin gallate (EGCG), presented higher anti-collagenase activity than that of both extracts ($p < 0.05$). Elastin can increase skin elasticity and prevent skin from sagging with age. If an extract has anti-elastase activity, it can maintain the skin's youthfulness and delay aging. This test was also a kinetic system similar to that used for the anti-collagenase activity. When comparing the extracts from *M. pruriens* var. *utilis* and *M. pruriens* var. *pruriens*, the extract from *M. pruriens* var. *utilis* had higher values. All of the results showed that the extracts from the two species possessed anti-aging activities and, thus, can be used as raw material in anti-aging products.

An occlusion assay was used to analyze the moisturizing properties of the *M. pruriens* seed extracts. This assay measured the ability of the extracts to prevent water loss or seal it in the skin. The results showed that the beakers sealed with the extracts had less water evaporation than the beakers filled with blank filter paper. The weight of the water that

evaporated was calculated as the occlusive factor (*F*). Occlusion occurs when the skin is directly or indirectly covered by impermeable materials, such as diapers, tape, chambers, gloves, textiles, garments, wound dressings, transdermal devices, and topical formulations containing fatty or oily substances. This creates an airtight barrier that traps moisture and promotes skin hydration and healing [37]. Occlusion can prevent water from escaping through diffusion, increases the moisture levels in the outer skin layer, causes skin cells to expand, and encourages the absorption of water into the spaces between skin cells. Fatty acids offer some benefits; occlusion provides a more practical approach to skin hydration by preventing water loss from the skin. In addition, the fatty acids did not directly affect the flexibility of the stratum corneum; instead, they indirectly enhanced the flexibility by preventing water loss [38]. In the analysis with GC–MS, fatty acids were found to be components of the seed extracts. The results of the occlusion test suggest that both varieties of *M. pruriens* seeds extracts were able to prevent the evaporation of water from the skin. Previous research showed that *M. pruriens* seeds contain a wealth of fatty acids, including polyunsaturated (linoleic and linolenic), monounsaturated (oleic), and saturated (myristic and palmitic) varieties [39]. These fatty acids are crucial components of three essential intercellular lipids that maintain the skin barrier function: sphingolipids, free sterols, and fatty acids. Naturally present in the skin's upper layer, fatty acids are vital in the barrier function and hydration. Long-chain fatty acids can effectively reduce moisture loss from the skin's surface [40]. Furthermore, they have an occlusive effect, protecting against excessive trans-epidermal water loss.

Jojoba oil has various benefits due to its skin moisturizing, antioxidant, and anti-aging properties. It has a function as an emollient, as it can add moisture to the skin by filling in the gaps between corneocytes. It can also help strengthen the skin and prevent water loss from it [41]. Therefore, it is a suitable oil for developing nanoemulsions. Nanoemulsions are colloidal dispersions consisting of oil, water, and surfactants. Water and oil are mixed with an appropriate surfactant. The droplet size of nanoemulsions is in the range of 20–500 nm. As a result, they can penetrate through rough skin and are stable with respect to precipitation, inherent creaming, flocculation, coalescence, and sedimentation due to their small droplet size. They can incorporate both hydrophilic and hydrophobic compounds. Nanoemulsions offer several advantages for the delivery of drugs, biological agents, and diagnostic agents. They can protect labile drugs, control drug release, increase drug solubility, increase bioavailability, reduce patient variability, and are non-toxic and non-irritating [27]. They can be used to prepare a variety of formulas such as foams, creams, liquids, and sprays. Moreover, they can control and deliver an active compound to a target site on the skin. However, the stability of nanoemulsions depends on their pH, temperature, and Ostwald ripening [10]. The improved skin penetration of nanoemulsions depends on the nano-sized range and composition [42]. Previous research found that nanoemulsions with particle sizes ranging between 20 and 62 nm are suitable for delivering active compounds through the skin [27]. The PDI values were less than 1, indicating good particle distribution (i.e., the particles did not clump together). Ultrasonication is a simple and effective method for breaking coarse emulsions down into smaller droplets using cavitation. Polysorbate 80 (Tween® 80) and sorbitan oleate (Span® 80) were used as emulsifiers in this study because they are non-toxic for the skin and have good efficacy when preparing nanoemulsions with a small particle size and narrow range of PDI values. In this study, unloaded nanoemulsions were developed with various concentrations of emulsifiers (5, 10, and 15 %*w*/*w*); the results showed that increasing the emulsifier concentration up to 10 %w/w reduced the particle size of the formulation, due to a decrease in the surface tension of the system. However, the use of an emulsifier concentration of 15 %*w*/*w* generated a particle size that was equal to that obtained with an emulsifier concentration of 5 %*w*/*w*. Therefore, it can be concluded that the optimal emulsifier concentration can generate the smallest droplet size in nanoemulsions. As mentioned above, the unloaded nanoemulsion consisting of emulsifiers at a concentration of 10 %*w*/*w* was chosen for the loading of the extract. The nanoemulsions containing *M. pruriens* var. *utilis* seed extracts

had a small particle size (less than 150 nm) in an acceptable size range with a low PDI, and they were stable over 30 days at room temperature (30 °C). The zeta potential values were also within an appropriate range and showed a negative charge. This negative charge helped repel the particles and prevented them from fusing. However, the particle size of the formulation slightly increased after six heating–cooling cycles because high temperatures affected the Oswald ripening of the particle size. The entrapment efficiency method can be used to evaluate the efficiency of the encapsulation in nanoemulsions. It was found that the nanoemulsions had a high entrapment efficiency. The choice of emulsifier in the formulation affected the degree of encapsulation. The HLB value is an important factor when encapsulating active ingredients in extracts. Polysorbate 80 (Tween® 80) and sorbitan oleate (Span® 80) have HLB values of 15 and 4.3, respectively. HLB values between 3 and 8 allow a bilayer surface to be prepared (water-in-oil, W/O emulsifier), while polysorbate 80 (Tween® 80) and sorbitan oleate (Span® 80) allow an oil-in-water emulsion to be prepared (O/W emulsifier) [43]. Additionally, high HLB values (>11) indicate hydrophilic (water-soluble) surfactants. They help increase the solubility of active ingredients in oil, and surfactants increase the entrapment efficiency. The research of Xu et al. (2018) [42] showed that nanoemulsions with vegetable oil and a synthetic surfactant were more effective at delivering to the skin. As shown by our results, nanoemulsions containing *M. pruriens* seed extracts can be suitable formulations for the further development of cosmeceutical products.

## 5. Conclusions

In summary, this study revealed that both *M. pruriens* var. *utilis* and *M. pruriens* var. *pruriens* offer numerous benefits as active ingredients in cosmetic formulations. A phytochemical analysis conducted using HPLC to identify L-dopa served as a reliable method for controlling the quality of *Mucuna pruriens* materials. Additionally, GC–MS analysis was employed to detect fatty acids in the extracts that were potentially associated with occlusive properties. The *Mucuna* species exhibited diverse phytochemicals with various biological activities. Within this investigation, *M. pruriens* var. *utilis* demonstrated superior antioxidant, anti-aging, and moisturizing properties to those of *M. pruriens* var. *pruriens*. This distinction was crucial when selecting an extract for incorporation into nanoemulsions. The nanoemulsions containing *M. pruriens* var. *utilis* extract exhibited characteristics such as a small particle size, low polydispersity index (PDI), and excellent stability. Furthermore, they displayed a high entrapment efficiency and efficient delivery of the active ingredient into the deeper layers of the skin. In conclusion, nanoemulsions containing *M. pruriens* var. *utilis* seed extract show promise for use in the development of new cosmeceutical products, and particularly for formulations targeting antiaging properties.

**Author Contributions:** Conceptualization, K.K. and A.I.; methodology, K.K., A.I. and T.T.; formal analysis, S.C., K.K., A.I. and T.T.; resources, K.K.; data curation, S.C., K.K., A.I. and T.T.; writing—original draft preparation, S.C. and K.K.; writing—review and editing, S.C., K.K., A.I. and T.T.; supervision, K.K.; project administration, K.K.; funding acquisition, K.K. and A.I. All authors have read and agreed to the published version of the manuscript.

**Funding:** This research was partially funded by the Faculty of Pharmacy, Chiang Mai University.

**Data Availability Statement:** Data are contained within the article.

**Acknowledgments:** The authors thankfully acknowledge the Faculty of Pharmacy at Chiang Mai University and the School of Cosmetic Science, Mae Fah Luang University, for providing us with the necessary resources.

**Conflicts of Interest:** The authors declare they have no conflicts of interest.

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
