# Peer review of "Nanoemulsions Containing Mucuna pruriens (L.) DC. Seed Extract for Cosmetic Applications"

_cosmetics, doi:10.3390/cosmetics11010029_

Round 1

Reviewer 1 Report

Comments and Suggestions for Authors

I have read with interest the manuscript of Chookiat and cols on a M. pruriens nanoemulsions formulations. The paper reports a study on the formulation, characterization and an in vitro skin retention study. The manuscript is complete with many studies such determination of phenolics and flavonoids content, antioxidant, antiaging activity and moisturizing properties.  The sudy and the characterization of the formulations developed is is well done and my suggestion is to publish this work after minor revisions, see below.

  1. The pH of skincare products plays a crucial role in determining their effectiveness. Topical products that have a pH that is too high or too low can be harsh on the skin and disrupt its natural pH balance, leading to dryness, irritation and inflammation. Do the autors consider necessary determination of pH? Explain it.
  2. In the formulations presented, surfactants, other excipients play a primary role. The systematic nomenclature in scientific terms is important. For this reason, I suggest to the authors to specify in the manuscript the systematic chemical names for the raw materials used in section 2 and all manuscript, and add company, city and country. For example describe Tween® 80 (polysorbate 80), Span® 80 (sorbitane monooleate),…
  3. Skin irritation studies are commonly performed for topically applied products.  Do the authors consider necessary to do it for the formulations developed? Explain it.

4.      Generally, topical application products require viscosity measurements. Have the authors considered determining the viscosity and flow of the nanoemulsions developed? Do the authors consider necessary to do it? Explain it.

5.      What is the temperature preparation of the formulations developed? Do the authors need heating for the preparation formulations? Please rewrite elaboration section (2.10.1)

  1. Determination by HPLC. It is important to add information on the validation parameters. Please rephrase and cite the complete analytical method used for the determination of l-Dopa and skin retention.

7.      Do the authors consider the stability of formulations developed? Do the authors perform short-term and/or large-term stability studies about formulations developed? Do the authors believe the stability at room temperature for 30 days is sufficient? Explain it.

8.      In section 3.8.1 the authors prepare 3 formulations (1,2,3) in which the concentrations of the surfactant vary and without identifying in the text and graphics which corresponds to each one. Please indicate each formulation to which concentration it corresponds. Furthermore, the authors would have to mention it in section 2.10.1

  1. The authors should describe more correctly all the equipment used in all manuscript. Please add: Commercial brand, model. And company, city and country in brackets.
  2. Use DLS to describe technology used to describe particle size (2.10.2)

11.  Line 303. What is Stat M membrane? Do the authors mean Strat® M membrane supllied by Merck Millipore? Please correct it

12.  Line 281. Describe suchapter (2.11.2) like 2.10.2 to improve understanding

13.  Line 330-331 describe quercetin correctly

14.  Line 133 must be 2.4 and succesivelly

  1. Change minutes for min in all manuscript

Author Response

Dear Reviewer,

We greatly appreciate the valuable comments and suggestions from the reviewer. We have carefully read and responded to all comments, point by point. The specific alterations in the manuscript in response to the reviewer's comments are shown in yellow highlights. In addition, the manuscript has been edited English grammar by MDPI and Mr. Thomas McManamon, Faculty of Pharmacy, Chiang Mai University, Thailand.

We hope all of the changes have addressed the reviewers’ concerns, so with these additions, we hope our work will be accepted for publication in Cosmetics.

Best regards,

Asst. Prof. Dr. Kanokwan Kiattisin

Reviewer 2 Report

Comments and Suggestions for Authors

1.      Source of each chemical must be mentioned  in Material section

2.      Short  forms of various chemicals used throughout the manuscript Which are difficult to understand  AAAVPN   EGCG  etc.

3.      In line 275, the stability of nano-emulsion was estimated at room temperature, which is undefined.

4.      Undersection 2.11.1, the title discloses loading of extract in nano-emulsion, however in the concerned para, it has been referred to preparation method of nano-emulsions only.

5.      Figures 2 and 3 can be combined.

6.      The HPLC method utilized for estimation of L-dopa, was neither referred from any standard reported method nor a developed method.

7.      Tailing in the HPLC chromatograms (fig 5) was observed, therefore, percentage impurities present must be reported.

8.      FTIR or NMR was not done, recommended for percentage impurity detection of the extract.

9.      Line no302  -  In this study   2 times written.

1-Formulation-optimization of nano-emulsion is not presented in appropriate manner i.e. selection basis of surfactant, oil phase and extract was not defined.

1- What were characteristic features of optimized formulation, and which formulation was characterized is unclear [variety of plant species is not mentioned properly] (lines 605-654)

-   -No SEM and TEM images and no Particle Size-Analyzer data is there to confirm the reported size, PDI and zeta-potential of nano-formulation.

1- Bar graphs of figure 17, can be presented in a single graph.

1- Draft is required to be thoroughly checked for grammatical-mistakes, repetition of words and editing-formatting errors.

Comments on the Quality of English Language

1  Quality of language is poor and needs major revision

Author Response

Dear Reviewer,

We greatly appreciate the valuable comments and suggestions from the reviewer. We have carefully read and responded to all comments, point by point. The specific alterations in the manuscript in response to the reviewer's comments are shown in green highlights. In addition, the manuscript has been edited for English grammar by MDPI and Mr. Thomas McManamon, Faculty of Pharmacy, Chiang Mai University, Thailand.

We hope all of the changes have addressed the reviewers’ concerns, so with these additions, we hope our work will be accepted for publication in Cosmetics.

Best regards,

Asst. Prof. Dr. Kanokwan Kiattisin

Round 2

Reviewer 2 Report

Comments and Suggestions for Authors

Paper can be accepted in the present form